# Changes in Psychological Outcomes after Cessation of Full Mu Agonist Long-Term Opioid Therapy for Chronic Pain

**DOI:** 10.3390/jcm12041354

**Published:** 2023-02-08

**Authors:** Marcelina Jasmine Silva, Zhanette Coffee, Chong Ho Alex Yu, Joshua Hu

**Affiliations:** 1The Focus on Opioid Transitions (FOOT Steps) Program, IPM Medical Group, Walnut Creek, CA 94598, USA; 2College of Nursing, University of Arizona, Tucson, AZ 85721, USA; 3Office of Institutional Research, Azusa Pacific University, Azusa, CA 91702, USA; 4College of Osteopathic Medicine, Touro University, Vallejo, CA 94592, USA

**Keywords:** chronic pain, opioids, patient experiences, psychological outcomes, management, treatment, buprenorphine

## Abstract

Improved understanding of psychological features associated with full mu agonist long-term opioid therapy (LTOT) cessation may offer advantages for clinicians. This preliminary study presents changes in psychological outcomes in patients with chronic, non-cancer pain (CNCP) after LTOT cessation via a 10-week multidisciplinary program which included treatment with buprenorphine. Paired *t*-tests pre- and post-LTOT cessation were compared in this retrospective cohort review of data from electronic medical records of 98 patients who successfully ceased LTOT between the dates of October 2017 to December 2019. Indicators of quality of life, depression, catastrophizing, and fear avoidance, as measured by the 36-Item Short Form Survey, the Patient Health Questionnaire-9-Item Scale, the Pain Catastrophizing Scale, and the Fear Avoidance Belief Questionnaires revealed significant improvement. Scores did not significantly improve for daytime sleepiness, generalized anxiety, and kinesiophobia, as measured by the Epworth Sleepiness Scale, the Generalized Anxiety Disorder 7-Item Scale, and the Tampa Scale of Kinesiophobia. The results suggest that successful LTOT cessation may be interconnected with improvements in specific psychological states.

## 1. Introduction

The medical community was called to action by The Centers for Disease Control and Prevention (CDC) 2016 guidelines to decrease exposure to full mu agonist long-term opioid therapy (LTOT) for chronic, non-cancer pain (CNCP) in an effort to stem the resulting sequelae of the increasing incidence of opioid overdose [1]. Less severe, but still disturbing, adverse full mu agonist opioid medication effects are numerous and well documented, ranging from immediate (cognitive impairment, dry mucous membranes, slowed intestinal motility) [2] to long-term and insidious (hypogonadism [3], immune compromise [4], and hyperalgesia) [5,6]. Associations between opioid use and declining mood states have also been well documented, such as a decline in patient-perceived quality of life [7], and increased measures of depression [8,9,10,11], catastrophizing [10,12,13], fear avoidance [14], and anxiety [10,12,15,16]. However, unforeseen complications regarding efforts to reduce or discontinue opioid dosing within the patient population that utilizes LTOT are becoming apparent [17,18], with recent reports documenting associated overdose and suicide [19,20,21]. Improved understanding of psychological features associated with a cohort of patients who successfully ceased LTOT may offer useful directives for how to proceed effectively when LTOT cessation is desired.

One potential alternative to LTOT is buprenorphine. Buprenorphine is an opioid drug with partial mu agonist properties [22]. It can be a safer option than full mu agonist opioids for some patients [23,24], as the dangers of respiratory depression and overdose-related death associated with opioid use are thought to be conferred by full mu receptor agonism [24,25]. However, partial mu agonism, in the case of buprenorphine, confers potent analgesia [26,27,28,29].

The primary aim of this study was to identify changes between psychological assessment questionnaires, pre and post LTOT cessation, for patients with CNCP. We present findings from a retrospective analysis in a cohort of patients with CNCP who were successfully able to cease LTOT through participation in a group, multidisciplinary program [14,30] that offered therapeutic options that included buprenorphine. This study took place prior to reports of declining clinical outcomes after LTOT cessation, which notably did not include the option of buprenorphine in care planning [2,3,4,5,6]. Informed by multiple reports of declining mood with opioid initiation [8,9,10,11,12,13,14,15,16], the researchers hypothesized that LTOT cessation would result in improved mood outcomes.

## 2. Materials and Methods

### 2.1. Study Design

De-identified data were collected via a retrospective review of electronic medical records (EMR) from October 2017 to December 2019, comparing the pre- and post-psychological assessment questionnaire scores of 98 patients with CNCP who successfully ceased LTOT use through participation in a previously described, group multidisciplinary program [14,30]. Questionnaires were given to each patient at orientation and at graduation. Pre- and post-LTOT cessation scores were paired and used for analysis in this study.

### 2.2. Intervention

The multidisciplinary program [14,30,31] operated as a stand-alone intervention within a larger, multi-center, private practice specializing in CNCP in Northern California. Two centers and clinical teams participated in program administration under one medical director. Patients in the program met for approximately six hours once a week for ten weeks. The standardized curriculum entailed group cognitive behavioral therapy, group home exercise training utilizing complimentary care activities, and individualized medication management. Buprenorphine was offered to each patient as an alternative to LTOT. Extended panel urine drug screening was mandated at each meeting to corroborate participant compliance.

### 2.3. Participants

Study participants were comprised of the 98 successful graduates of a multidisciplinary LTOT cessation program that commenced between October of 2017 and December of 2019. A total of 109 patients started the program, and 11 either voluntarily left or were referred to a higher level of care due to high acuity comorbidities diagnosed after admittance. Program inclusion criteria were: adult-aged patients who voluntarily enrolled for the purpose of LTOT cessation due to lack of satisfaction with pain control, medication effects, and/or functional capacity while on LTOT. Participants were diagnosed with CNCP from any etiology; had used daily LTOT at the time of admission for a minimum of a year’s duration (although most reported much longer use), or struggled to maintain recent opioid cessation after prolonged use; had previously tried and failed or plateaued in regards to opioid weaning. Program recruitment, exclusion criteria, and criteria for referral have been described elsewhere [14,30,31]. Long-term opioid therapy was defined as any form of prescribed oral or transdermal long- or short-acting pharmaceutical opioid obtained while under the care of a physician. At admission, participants used LTOT amounts as high as 600 daily oral morphine milligram equivalents (MME) (median, 60 MME; 25% quartile, 36.5 MME; 75% quartile, 90 MME; interquartile range, 53.5 MME) [30].

### 2.4. Measures

Standardized psychological questionnaires were given to each patient at the program orientation meeting and again at the program graduation. Each questionnaire was previously and independently validated. Table 1 lists and describes the application of the Epworth Sleepiness Scale (ESS) [32], the 36-Item Short Form Survey (SF-36) [33], the Generalized Anxiety Disorder–7 Item Scale (GAD-7) [34], the Patient Health Questionnaire (PHQ) [35], the Fear Avoidance Belief Questionnaire-Physical Activity (FAB-PA) and Work (FAB-W) [36], the Tampa Kinesiophobia Scale (TSK) [37], the Pain Catastrophizing Scale (PCS) [38], and the Brief Pain Inventory-Pain Severity (BPI-Pain) and Impairment (BPI-Impairment) survey [39].

Of note, additional data were gathered throughout the program for the separate analyses of protective and hindering psychological and clinical features associated with LTOT cessation success, and these can be found in earlier publications [14,30].

### 2.5. Analysis

Descriptive statistics were derived from retrospective data found in the demographic section of the EMR for program graduates. Paired *t*-tests were used to compare dataset means from psychological questionnaires pre- and post-LTOT cessation. Prior to paired comparison, principal component analysis (PCA) was used to compare individual SF-36 subcategory results for the pre- and posttests. The SF-36 is typically reported as 8 individual scores—one for each category. However, since running multiple *t*-tests for all SF-36 subcategories might inflate the Type I error rate and result in false findings, dimension reduction was employed to minimize the number of variables to be tested. In PCA, data visualization techniques, including the loading plot [44,45] and the scree plot, were utilized in order to identify the proper number of principal components.

## 3. Results

### 3.1. Descriptive Statistics

The study began with 109 clinical program participants; 11 left the clinical program and were lost to the follow-up. Thus, 98 subjects (representing 90% of the program participants) remained under observation after the successful cessation of LTOT during the 10-week program [30] and were included in the present study. Of those who successfully graduated, 95 subjects (97%) chose to use buprenorphine as an LTOT cessation tool. Program participants were 27 to 88 years old; 69% were identified as female. A broad range of payer sources were noted: 30% Medicare, 25% industrial insurance, 10% Medicaid, 44% commercial insurance, and <1% no insurance.

### 3.2. Principal Component Analysis and Scree Plot for the SF-36

The reliability index, as measured by the standardized Cronbach’s alpha of the pretest of SF-36 is 0.8899, compared with 0.846 at the posttest, implying that the response patterns to these questions are internally consistent. In the loading plot, the axes of the plots are the principal components, while the variables are symbolized by vectors radiating from the center. Vectors pointing in the same direction, with a small angle between them, implies a positive and close relationship between the variables. Due to this characteristic, they could be loaded into the same component. Figure 1a indicates that all observed variables in the pretest of SF-36, as depicted by the vectors, pointed in the same direction and are close to each other. Figure 1b suggests that the observed items in the posttest of SF-36 could be classified into two groups, based on the clustering patterns.

In the scree plot (Figure 2a,b), the y-axis represents the eigenvalue, which is the sum of the squares of the loadings, whereas the x-axis denotes the number of potential components. Although the loading plot of the posttest indicates a 2-component model, both scree plots suggest that one single component is sufficient to yield the highest eigenvalue. Considering this finding, the average pretest and posttest scores of all SF-36 items were used for paired *t*-tests.

### 3.3. Paired t-Tests

Paired *t*-tests were used to compare dataset means from psychological questionnaires pre- and post-LTOT cessation (Table 2). As indicated in the table, six out of ten paired *t*-tests yielded a significant *p*-value (<0.0001). Because multiple *t*-tests were conducted, the Type 1 error rate might be inflated. The Bonferroni correction rectifies the situation by dividing the alpha level by the number of tests. In this case, the adjusted alpha level is 0.05/10 = 0.005. Nevertheless, because the *p*-value of all significant results is <0.0001, which is far lower than 0.005, the conclusion remains the same.

Data imputation was considered for this study for the inconsistent n value, but was ruled out because some participants deliberately chose to skip several questions or entire assessments. When data are missing due to inherent lack of randomness, it is impossible to account for systematic differences between the missing and the observed values based on the existing data, especially when the reference data are sparse. Further, when all or many responses in a group of questions are missing, multiple imputation will yield misleading results [46].

## 4. Discussion

### 4.1. Improved Psychological Assessment Scores Post-LTOT Cessation

This study found that the improvement between the pre- and post-cessation indicators of quality of life, depression, pain interference, catastrophizing, and fear avoidance (assessed via the SF-36, PHQ, BPI-Pain, BPI-Impairment, PCS, and FAB-PA, respectively) among subjects with CNCP who successfully ceased LTOT are significant at the level of *p* < 0.00l. These findings are consistent with those in previous studies examining mood changes affiliated with problematic opioid use, which have described an intercorrelation between such use and increased depression [8,9,10,11,47], fear avoidance [43], catastrophizing [10,12,13,38,48], and decreased quality of life [7] with increased states of pain [12,14,15,16,47,48,49,50,51,52,53,54,55,56,57,58,59,60]. Interpretation of the present findings must consider not only the impact of LTOT cessation, but also that of buprenorphine introduction and concurrent participation in a group multidisciplinary program, or even a potential synergy between these factors.

Most subjects utilized buprenorphine for LTOT cessation. Buprenorphine acts as a partial mu agonist and as an antagonist at the delta [26,55] and kappa [24,26,55] opioid receptors [26,57,58,59]. These receptor dynamics have been theorized to confer anxiolytic [56] and anti-depressive effects [50,58,59,60,61,62,63,64] in opioid-dependent [56] and opioid-naïve [47,55,58,60] patients and to improve patient-reported scores regarding quality of life [28,50]. Perhaps the addition of buprenorphine can explain, at least in part, the discrepancy between the participants’ mood improvements compared with recent studies documenting clinical deterioration with LTOT cessation [19,20,21]. This dynamic would be an interesting area of future study.

The effects of both partial and full mu agonist opioid exposure on quality of life have been previously studied. One study showed that full mu agonist opioid exposure lowered the SF-36 score initially, but no difference was found between patients with CNCP exposed to opioids vs. not exposed after 4 years duration [7]. Another reported that opioid-naïve patients exhibited improvements in SF-36 scores after 8 weeks of daily treatment with buprenorphine for the indication of treatment-resistant depression [50]. Similarly, a different study found that switching from LTOT to sublingual buprenorphine for at least 60 days resulted in improved quality of life scales, as measured using the QOLS [28]. Longer term follow-up studies would be helpful to better understand the relationship between LTOT, buprenorphine, and quality of life.

In the current dataset, the post-LTOT cessation depression scores (assessed via the PHQ) show a mean improvement from moderately severe to moderate depression, which trends consistently with the results in previous publications. The link between depression and problematic full mu agonist opioid use has been previously documented from a variety of angles. Higher depression scores have been correlated with opioid misuse in CNCP patients with no prior substance use disorder [10]. Depression has also been identified as a risk factor for prolonged post-surgical opioid use [11]. Similarly, researchers have found that opioid use is a risk factor for depression, independent of pain [8,9]. Buprenorphine’s effect on depression has also been studied in patients with and without concurrent opioid use disorder and was shown to provide marked improvement in suicidal ideation and treatment-resistant depression after administration [47,58]. Similarly, in a meta-analysis conducted by Serafini et al., 13 studies concluded that buprenorphine alone, or in co-administration with other opioid antagonists, may significantly reduce depression [59]. The current data, supported by previous studies, strongly suggest that depression is an important target for assessment and therapy in the pursuit of LTOT cessation.

Previous reports regarding the PCS and FAB are nuanced. For the PCS, it is interesting to note that a score of 30 or higher was initially validated as clinically relevant [38,42]. However, lower scores have been documented to be associated with the chronicity of prolonged recovery and delayed return to work [45]. Therefore, the substantial improvement in the PCS score with LTOT cessation seen in this study can be argued to be clinically significant, despite being below 30 on the pretest, especially considering the scale of the improvement. Targeted psychosocial therapy to improve catastrophizing has been shown to significantly aid in returning to work after a period of disability [48]. Current results suggest that addressing catastrophizing may be an avenue worthy of future study for promoting LTOT cessation for patients with CNCP.

The optimal cut off for determining significant FAB scores has been studied in several contexts and varies respectively [36,40,61,62,63,64]. Higher FAB scores have been correlated with an increased probability of current and future work loss and disability [36,65,66], as well as social withdrawal [67] and increased LTOT reliance [14]. FAB analysis may also help determine which clinical interventions have an increased probability of a successful outcome to decrease patient-reported disability and pain [36,68,69]. Of note, some of the utility of the FAB, when correlated in these specific ways, has been validated only in industrially insured patients [69], and the FAB-W has been validated for currently—or recently—working patients [36,68,70]. Thus, the present FAB results are difficult to analyze based on previous validations, as the current study did not exclude participants who identified as disabled or retired, nor did it control for insurance payer type.

The BPI scores warrant notice in that their improvement implies a general lack of suffering among the participants, despite LTOT cessation.

### 4.2. Psychological Assessment Scores Lacking Significant Improvement Post-LTOT Cessation

Equally interesting is the lack of association seen between LTOT cessation and daytime sleepiness, generalized anxiety, and kinesiophobia (assessed via the ESS, GAD, and TSK, respectively). Nighttime sleep disturbance and daytime impaired cognition are common anecdotal complaints among patients on LTOT. Disturbed sleep architecture and increased obstructive and central sleep apnea have long been recognized as side effects of chronic opioid use [70,71]. Thus, it was surprising that LTOT cessation did not pair with improved daytime sleepiness. However, this outcome is consistent with previous studies documenting that sleep disturbance does not necessarily correlate with daytime sleepiness, as measured by the ESS, in chronic opioid users [70,71,72].

The improvement in anxiety (assessed via the GAD) between pre- and post-LTOT cessation was not significant. Previous studies have suggested that a self-medication model has been implicated in inciting, or exacerbating, opioid use disorder for patients with clinically significant anxiety [12,16,51], and higher anxiety scores have also been implicated in the increased length of post-operative opioid analgesic use [10]. Moreover, studies have found buprenorphine administration to be correlated with improvement in anxiety [47,56]. The relationship between anxiety and LTOT reliance warrants increased study to help reconcile present and previous findings.

Kinesiophobia (TSK) scores did not follow the significant improvement trajectory of the PCS and FAB- PA. These tools are often grouped together in the category of anxiety, or fear-based, belief and behavior assessments. However, previous investigation into their interchangeability has failed to show cross-over reliability [73]. Of note is the fact that many participants chose not to complete the TSK, for unknown reasons, and it was by far the most neglected of the tests by the participants, with an “n” of 34. Perhaps the smaller number of datasets affected the significance of this outcome.

### 4.3. Contribution to the Field

The present findings help to identify psychological improvements associated with successful LTOT cessation. While this study does not prove whether the intervention of the medication changes improved the mood outcomes, or whether the multidisciplinary program promoted a foundation of mood change that allowed for LTOT cessation, it is clear that specific mood improvements were associated with LTOT cessation. With further study, this dynamic may eventually suggest inroads to safer and more optimal treatments in the field of CNCP. Just as previous studies have shown that addressing high fear avoidance and catastrophizing beliefs improved disability measures [48,74], the current findings could provide the foundation to target specific psychological states for education and multidisciplinary support to aid in successful LTOT cessation in future trials.

### 4.4. Study Limitations

Studies that increase understanding of the clinical features that promote LTOT cessation are timely, yet scant, thus making the present findings relevant, despite the following design limitations: the data for this study were gathered retrospectively from a moderately-sized treatment cohort of non-randomized, non-blinded patients, who were treated in a private practice setting, and demographic subject data is limited. The lack of a long-term follow-up of mood status is also a limitation, and this follow-up would be especially interesting here, as an affiliated study showed prolonged cessation of LTOT in the current group [30].

## 5. Conclusions

This study found that improvements in indicators of quality of life, depression, pain interference, catastrophizing, and fear avoidance are associated with successful LTOT cessation in patients with chronic, non-cancer pain who participated in a multidisciplinary opioid cessation program that offered buprenorphine as a substitution. Further research is required to confirm the nature of this relationship. A better understanding of these potentially interdependent clinical phenomena may eventually help clarify and improve interventions for the cessation of LTOT in patients with CNCP.

## Figures and Tables

**Figure 1 jcm-12-01354-f001:**
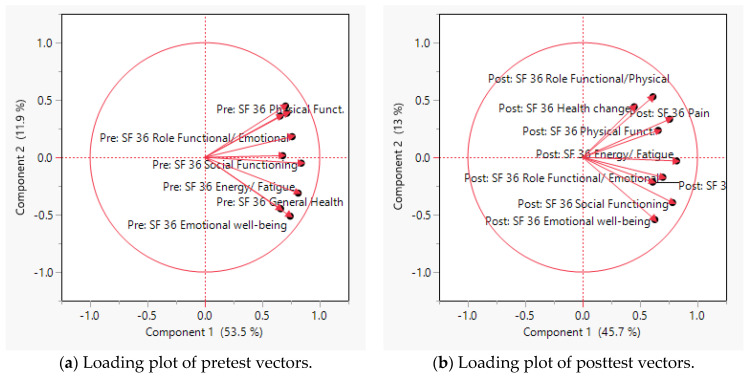
Loading plots of SF-36 vectors.

**Figure 2 jcm-12-01354-f002:**
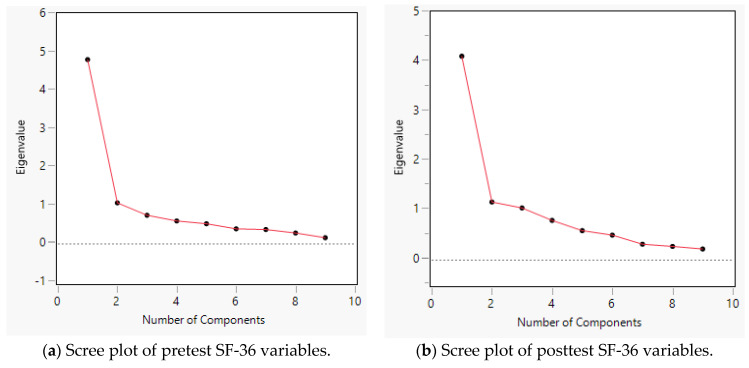
Scree plot of SF-36 variables.

**Table 1 jcm-12-01354-t001:** Psychological assessment questionnaires.

Questionnaire	Application
Epworth Sleepiness Scale (ESS) [32]	Measures daytime sleepiness: 0 to 10 is normal; 11–12 is mild; 13–15 is moderate excessive; and 16–24 is severe excessive.
The 36-Item Short Form Survey (SF-36) [33]	A total of 8 categories are scored on a scale from 0 to 100, with 100 representing the highest level of functioning possible: physical functioning, bodily pain, role limitations due to physical health problems, role limitations due to personal or emotional problems, emotional well-being, social functioning, energy/fatigue, and general health perceptions.
Generalized Anxiety Disorder 7-Item Scale (GAD-7) [34]	Assesses anxiety: 0–5 is mild, 6–10 is moderate, and 11–15 is high.
Patient Health Questionnaire (PHQ) [35]	Assesses depression: 1–4 is minimal, 5–9 is mild, 10–14 is moderate, 15–19 is moderately severe, and 20–27 is severe.
Fear Avoidance Beliefs Questionnaire—Work and Physical Activity (FAB-Wand PA) [14,36,40]	Two subscales (FAB-W: 0-42; FAB-PA 0-24) in which higher scores indicate more severe pain and disability due to fear avoidance beliefs about work and physical activity. Various score thresholds have been documented as associated with clinical relevancy and specific negative chronicity of CNCP. Higher scores have been associated with poor physical and manual therapy results and low return to work rates after an injury.
Tampa Scale of Kinesiophobia (TKS) [37]	A measure of fear of movement and reinjury. Scores range from 17–68, with higher scores being of higher severity.
Pain Catastrophizing Scale (PCS) [14,38,41,42]	Assesses levels of catastrophizing. In initial validation, a score of 30 or more correlated with high unemployment, self-declared “total” disability, and clinical depression. However, various lower score thresholds have been documented as associated with clinical relevancy for specific negative chronicity of CNCP.
Brief Pain Inventory-Severity and Impairment (BPI-Pain and Impairment) [39,43]	Provides two scores which assess the severity of pain and pain-related impairment on daily functions using the mean of several Likert scales of 0–10, with 10 being the worst.

**Table 2 jcm-12-01354-t002:** Paired *t*-tests of pre- and post-LTOT cessation psychological assessment questionnaires.

Variable	Pretest Mean	Posttest Mean	Std. Error	n	t-Ratio	*p*	Lower 95% CI	Upper 95% CI
SF-36	36.95	49.76	2.41	64	5.31	<0.0001 *	7.99	17.62
ESS	8.08	8.47	0.46	61	0.84	0.4058	−0.54	1.31
PHQ	11.82	7.91	0.81	57	−4.80	<0.0001 *	−5.53	−2.29
GAD7	7.71	7.43	0.94	58	−0.29	0.7692	−2.15	1.60
BPI-Pain	5.91	4.67	0.20	58	−6.19	<0.0001 *	−1.65	−0.84
BPI-Impairment	6.35	4.56	0.30	60	−5.88	<0.0001 *	−2.39	−1.18
PCS	21.6	13.32	1.39	56	−5.92	<0.0001 *	−11.09	−5.48
FAB-PA	12.95	11.18	1.03	55	−1.71	<0.0001 *	−3.83	0.31
FAB-W	24.02	23.8	1.31	45	−0.14	0.8867	−3.01	1.39
TSK	35.72	35.16	1.42	34	0.39	0.6967	−3.39	2.29

* Significant at the 0.01 Alpha level.

## Data Availability

The de-identified data can be retrieved from the authors, upon request.

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
