# Peer review of "Changes in Psychological Outcomes after Cessation of Full Mu Agonist Long-Term Opioid Therapy for Chronic Pain"

_jcm, 2023, doi:10.3390/jcm12041354_

Round 1
Reviewer 1 Report
Journal of Clinical Medicine
Special Issue: Chronic Pain: Clinical Updates and Perspectives
Ref. No.: jcm-2108445
Title: Psychological Assessment Changes After Cessation of Mu Agonist Long Term Opioid Therapy for Chronic Pain
Overview and general recommendation:
Thank you for the opportunity to review this empirical study examining changes in psychological outcomes from pre- to post-cessation of mu agonist long term opioid therapy (LTOT). This paper is cohesive and importantly contributes to our understanding of clinical and psychological changes following opioid cessation. I have provided some questions for clarification and feedback for improvement below.
Title: The current phrasing is a bit confusing. I might consider changing the title to something like “Changes in psychological functioning after cessation of mu agonist long term opioid therapy for chronic pain.”
Abstract: Avoid using acronyms without defining them (e.g., CNCP). Similar to the title, consider rephrasing (e.g., “…presents changes in psychological functioning…” or “psychological outcomes”). It would also be helpful to include the timeframe (i.e., pre-cessation compared to how long post-cessation?). Edit for awkward phrasing (e.g., “psychological assessment scoring of quality of life,… revealed significant improvement” could be changed to “There were significant improvements in quality of life, depression,…from pre- to post-cessation”). To consolidate, consider removing “as measured by the…” and instead: “…quality of life (SF-36), depression (PHQ-9), catastrophizing (PCS),…”—you can use the full names of questionnaires rather than abbreviations.
General comment: Typically, past tense (not present) is used when describing study aims, methods, and findings.
Introduction:
Overall, the introduction is concise and cohesive.
1. It could be helpful to provide some examples of which clinical outcomes worsen after LTOT cessation.
2. I recommend ending the introduction with aims and hypotheses. The paragraph about buprenorphine could be moved up, and the aims of this study moved down. What research has already been done in this area? Based on the literature, what were the authors’ hypotheses regarding change in psychological outcomes post-cessation?
Methods:
1. How many total patients completed the program between Oct 2017 and December 2019? While 98 patients met criteria and were included in this study, how many were excluded (i.e., were not able to successfully graduate from the LTOT cessation program).
2. Measures: what is meant by “significance” of the measures? In Table 1, validity is only discussed for the PCS. Some of the measures (FAB, PCS) are also missing descriptive information about the scale ranges.
Results:
1. I see you included total number of participants here. If 109 total patients started the program and 10 left, there would be 99, not 98, patients remaining—what happened to the 1 missing patient?
2. What was the average age of participants? Do the authors have any information about racial/ethnic breakdown of the sample?
3. Per Table 2, there appears to be a lot of missing data (n’s range from 34 to 64 across variables). What is the reason for the missing data? Were there any efforts to impute missing data?
4. Given the focus of this article on examining relationships between LTOT cessation and psychological outcomes, did the authors consider conducting correlation analyses to examine these relationships more specifically (e.g., relationships between measures of opioid use or misuse, such as the COMM, and psychological constructs)?
Discussion:
1. Avoid questionnaire acronyms—instead: “…improvements in quality of life, depressive symptoms, pain intensity, etc.”
2. The authors do a nice job of discussing psychological changes in the context of buprenorphine introduction (in addition to LTOT cessation). A future direction they could propose would be to compare those who use vs. do not use buprenorphine during LTOT tapering to determine whether improvements are due to LTOT cessation or buprenorphine (it would not be possible to assess in this sample as 97% of patients used buprenorphine).
3. The relationships between LTOT cessation and psychological outcomes need to be contextualized—the pre-post improvements are not solely related to LTOT cessation, but primarily to participating in a multidisciplinary program for chronic pain. That is, improvements in psychological constructs are not necessarily reflective of discontinuing LTOT but of a multimodal treatment approach. This needs to be emphasized. To truly understand the psychological characteristics associated with LTOT cessation specifically, randomized trials are needed—particularly ones that include LTOT cessation in the absence of a multimodal intervention. Discussion of improvements in PCS, FAB, and BPI should be modified accordingly. For example, this multimodal intervention may improve catastrophizing, fear avoidance beliefs, and pain, which may be beneficial for LTOT cessation; or it is possible that LTOT cessation contributes to decreased pain and unhelpful cognitions. The authors do not discuss some of the other variables that were improved (e.g., quality of life and depression).
4. I would discourage the authors from using the term “correlate” as correlation analyses were not conducted (e.g., “it was surprising that LTOT cessation did not correlate with improved ESS scores").
5. Any speculation as to why TSK scores did not improve?
6. Limitations: “Data were” not “data was.” Another limitation worth noting is the lack of long-term follow-up. This study only assesses psychological functioning in the acute cessation period.
7. Conclusions: As stated above, avoid questionnaire acronyms—instead: “…improvements in quality of life, depressive symptoms, pain intensity, etc.” Correct awkward phrasing (e.g., “in the setting of CNCP” could be changed to “in patients with chronic non-cancer pain.”
Author Response
Thank you for the opportunity to revise our manuscript. Please refer to the uploaded document for responses to both reviewers in a single document.
(A note to the editor from the authors: Reviewer #1 was exceptionally thoughtful and detailed in their contributions in all sections of the review. Please pass on our compliments and gratitude for their queries and their efforts to help us clarify our research and make it more relevant.)
Reviewer #1
Journal of Clinical Medicine
Special Issue: Chronic Pain: Clinical Updates and Perspectives
Ref. No.: jcm-2108445
Title: Psychological Assessment Changes After Cessation of Mu Agonist Long Term Opioid Therapy for Chronic Pain
Overview and general recommendation:
Thank you for the opportunity to review this empirical study examining changes in psychological outcomes from pre- to post-cessation of mu agonist long term opioid therapy (LTOT). This paper is cohesive and importantly contributes to our understanding of clinical and psychological changes following opioid cessation. I have provided some questions for clarification and feedback for improvement below.
Title: The current phrasing is a bit confusing. I might consider changing the title to something like “Changes in psychological functioning after cessation of mu agonist long term opioid therapy for chronic pain.”
The writers appreciate this critique, however, there is some variability between outcomes of the psychological assessments utilized – even when they assess similar phenomena. The term “psychological functioning” may be overreaching. Perhaps “Changes in Psychological Outcomes…” would be satisfactory.
Abstract: Avoid using acronyms without defining them (e.g., CNCP). …
This has been corrected.
…Similar to the title, consider rephrasing (e.g., “…presents changes in psychological functioning…” or “psychological outcomes”)….
This has been corrected.
… It would also be helpful to include the timeframe (i.e., pre-cessation compared to how long post-cessation?). …
Timeframe has been included.
… Edit for awkward phrasing (e.g., “psychological assessment scoring of quality of life,… revealed significant improvement” could be changed to “There were significant improvements in quality of life, depression,…from pre- to post-cessation”). To consolidate, consider removing “as measured by the…” and instead: “…quality of life (SF-36), depression (PHQ-9), catastrophizing (PCS),…”—you can use the full names of questionnaires rather than abbreviations.
This section was edited for phrasing.
General comment: Typically, past tense (not present) is used when describing study aims, methods, and findings.
Edits were made to attempt to achieve this throughout the text.
Introduction:
Overall, the introduction is concise and cohesive.
- It could be helpful to provide some examples of which clinical outcomes worsen after LTOT cessation.
This suggestion was implemented.
- I recommend ending the introduction with aims and hypotheses. The paragraph about buprenorphine could be moved up, and the aims of this study moved down. What research has already been done in this area? Based on the literature, what were the authors’ hypotheses regarding change in psychological outcomes post-cessation?
This suggestion was implemented.
Methods:
- How many total patients completed the program between Oct 2017 and December 2019? While 98 patients met criteria and were included in this study, how many were excluded (i.e., were not able to successfully graduate from the LTOT cessation program).
Edits were made to make this information more obvious.
- Measures: what is meant by “significance” of the measures? In Table 1, validity is only discussed for the PCS. Some of the measures (FAB, PCS) are also missing descriptive information about the scale ranges.
“Significance & Validation” was changed to “Application” in an attempt to address this query.
Descriptive information was added for the FAB and PCS, per the critique
Results:
- I see you included total number of participants here. If 109 total patients started the program and 10 left, there would be 99, not 98, patients remaining—what happened to the 1 missing patient?
Thank you for pointing out this typographical error. It was 11 who did not complete the program. This error was corrected.
- What was the average age of participants? Do the authors have any information about racial/ethnic breakdown of the sample?
Unfortunately, race was not uniformly noted in the EMR, and was not recoverable upon retrospective review. Also, the data was not collected in a way to gather the average age, although the youngest and oldest ages were noted.
- Per Table 2, there appears to be a lot of missing data (n’s range from 34 to 64 across variables). What is the reason for the missing data? Were there any efforts to impute missing data?
This critique was addressed via an addition to the manuscript. ”Data imputation was considered for this study for the inconsistent n value, but was ruled out because some participants deliberately chose to skip several questions or entire assessments. When data are missing due to inherent lack of randomness, it is impossible to account for systematic differences between the missing and the observed values based on the existing data, especially when the reference data are sparse. Further, when all or many responses in a group of questions are missing, multiple imputation will yield misleading results[1]”
- Given the focus of this article on examining relationships between LTOT cessation and psychological outcomes, did the authors consider conducting correlation analyses to examine these relationships more specifically (e.g., relationships between measures of opioid use or misuse, such as the COMM, and psychological constructs)?
Yes, these additional measures were the focus of earlier publications exploring positive and negative psychological indicators for which participants were successful to cease LTOT, which were used as references for the present study. A statement indicating this was added to section 2.4: “Of note, additional data were gathered throughout the program for a separate analysis of protective and hindering psychological and clinical features associated with program success and can be found in earlier publications[2], [3].”
Discussion:
- Avoid questionnaire acronyms—instead: “…improvements in quality of life, depressive symptoms, pain intensity, etc.”
This suggestion was implemented
- The authors do a nice job of discussing psychological changes in the context of buprenorphine introduction (in addition to LTOT cessation). A future direction they could propose would be to compare those who use vs. do not use buprenorphine during LTOT tapering to determine whether improvements are due to LTOT cessation or buprenorphine (it would not be possible to assess in this sample as 97% of patients used buprenorphine).
This statement was added to address this point: “Perhaps the addition of buprenorphine can explain, at least in part, the discrepancy between the participants’ mood improvements compared with recent studies documenting mood deterioration with LTOT cessation [citations added in text]. This dynamic would be an interesting area of future study. “
- The relationships between LTOT cessation and psychological outcomes need to be contextualized—the pre-post improvements are not solely related to LTOT cessation, but primarily to participating in a multidisciplinary program for chronic pain. That is, improvements in psychological constructs are not necessarily reflective of discontinuing LTOT but of a multimodal treatment approach. This needs to be emphasized. To truly understand the psychological characteristics associated with LTOT cessation specifically, randomized trials are needed—particularly ones that include LTOT cessation in the absence of a multimodal intervention. Discussion of improvements in PCS, FAB, and BPI should be modified accordingly. For example, this multimodal intervention may improve catastrophizing, fear avoidance beliefs, and pain, which may be beneficial for LTOT cessation; or it is possible that LTOT cessation contributes to decreased pain and unhelpful cognitions. The authors do not discuss some of the other variables that were improved (e.g., quality of life and depression).
These helpful clarification points were addressed via extensive edits to the first paragraph in section 4.1 and within section 4.3
- I would discourage the authors from using the term “correlate” as correlation analyses were not conducted (e.g., “it was surprising that LTOT cessation did not correlate with improved ESS scores").
The word “Correlate” was replaced in contexts referring to the current data, per the suggestion.
- Any speculation as to why TSK scores did not improve?
The authors hesitate to overreach with speculations in the manuscript, other than to note that this phenomena of lack of statistical improvement with the TSK while seeing improvement in PCS and FAB has been previously documented[4]. In this review response, we will speculate that all of these assessments seem to have specific applications for which they are best suited, both in terms of assessment and prediction for future outcomes. We discuss this possibility in a previous publication.[3] This critique did inspire us to add the following statement to the discussion: “Of note is the fact that many participants chose not to complete the TSK, for unknown reasons, and it was the most neglected of the tests by the participants by far with an “n” of 34. Perhaps the smaller number of data sets affected the significance of the outcomes here.”
- Limitations: “Data were” not “data was.” Another limitation worth noting is the lack of long-term follow-up. This study only assesses psychological functioning in the acute cessation period.
Grammatical change was made and the limitation noted.
- Conclusions: As stated above, avoid questionnaire acronyms—instead: “…improvements in quality of life, depressive symptoms, pain intensity, etc.” Correct awkward phrasing (e.g., “in the setting of CNCP” could be changed to “in patients with chronic non-cancer pain.”
This suggestion was implemented

Reviewer 2 Report
This manuscript sought to identify changes in psychological states pre- versus post-opioid cessation in persons with a history of long-term opioid therapy who participated in an opioid cessation program that included buprenorphine administration.
Overall, the manuscript has value in that there is little research in understanding the effects of opioid cessation, particularly as recent research has highlighted the potential for significant adverse consequences. While I have no overall issue with the data, the report itself could use some elaboration in certain areas, and the introduction and discussion do not seem aligned with one another.
1. The introduction feels overly brief. More information could be provided on the “worsening clinical outcomes” associated with LTOT, and a more robust rationale should be provided on the relationship between psychological states and LTOT, including its cessation. The authors note “improved understanding of psychological features associated with LTOT cessation ay offer advantages for clinicians.” But there is no context as to why the authors believe this, and what said advantages could potentially be.
2. More explanation on opioid cessation should be provided. What was the average daily MME of those engaged with the program? What was the average length of time participants were on LTOT prior to the intervention? Was cessation abrupt, with buprenorphine prescribed on day one of the intervention, or was there a taper? Was the dosage of buprenorphine the same for everyone or was this variable?
3. Could the authors clarify the timeline for the pre- and post- surveys. Were these the same for all individuals, or was it variable? For instance, was everybody provided the post-survey on the last meeting day of the 10-week intervention?
4. The results section needs some text for Table 2 rather than just one sentence that references Table 2.
5. The discussion reads more like an understanding of the utility of using certain psychological assessments to monitor the effects of opioid cessation rather than understanding psychological state changes post-opioid cessation. This is an interesting topic itself, but does not align with the introduction. There is some muddling of the research question and purpose of the study. The authors may consider a more concrete link between the introduction and discussion. If the authors wish to focus more on the assessment component, I think that is fine, but that should be reflected in the introduction.
Author Response
Thank you for the opportunity to revise our manuscript. Please refer to the uploaded document for responses to both reviewers in a single document.
Reviewer #2
This manuscript sought to identify changes in psychological states pre- versus post-opioid cessation in persons with a history of long-term opioid therapy who participated in an opioid cessation program that included buprenorphine administration.
Overall, the manuscript has value in that there is little research in understanding the effects of opioid cessation, particularly as recent research has highlighted the potential for significant adverse consequences. While I have no overall issue with the data, the report itself could use some elaboration in certain areas, and the introduction and discussion do not seem aligned with one another.
Major edits were undertaken to add the requested elaborations and better align the introduction and discussion.
- The introduction feels overly brief. More information could be provided on the “worsening clinical outcomes” associated with LTOT, and a more robust rationale should be provided on the relationship between psychological states and LTOT, including its cessation. The authors note “improved understanding of psychological features associated with LTOT cessation ay offer advantages for clinicians.” But there is no context as to why the authors believe this, and what said advantages could potentially be.
Edits and additions were made to address these critiques
- More explanation on opioid cessation should be provided. What was the average daily MME of those engaged with the program? What was the average length of time participants were on LTOT prior to the intervention? Was cessation abrupt, with buprenorphine prescribed on day one of the intervention, or was there a taper? Was the dosage of buprenorphine the same for everyone or was this variable?
Each of these queries are answered and described in detail in previous publications, including the medication protocols used [2], [3], [5], which are referenced in the materials and methods section. The authors defer to the editors to direct how much of these previous publications should be redescribed here, as requested by the reviewer. However, we agree that improved context is provided with the addition of MME amounts and a note on the minimum amount of time the participants used LTOT prior to the study. These changes were implemented.
- Could the authors clarify the timeline for the pre- and post- surveys. Were these the same for all individuals, or was it variable? For instance, was everybody provided the post-survey on the last meeting day of the 10-week intervention?
Edits were made to clarify this.
- The results section needs some text for Table 2 rather than just one sentence that references Table 2.
The following was added to address this critique: “Paired t-tests were used to compare data set means from psychological questionnaires pre and post LTOT cessation (Table 2). As indicated in the table, six out of ten paired t-tests yielded a significant p-value (<.0001). Because multiple t-tests were conducted, the Type 1 error rate might be inflated. The Bonferroni correction rectifies the situation by dividing the alpha level by the number of tests. In this case, the adjusted alpha level is .05/10=.005. Nevertheless, because the p-value of all significant results is <.0001, which is far lower than .005, the conclusion remains the same. “
- The discussion reads more like an understanding of the utility of using certain psychological assessments to monitor the effects of opioid cessation rather than understanding psychological state changes post-opioid cessation. This is an interesting topic itself, but does not align with the introduction. There is some muddling of the research question and purpose of the study. The authors may consider a more concrete link between the introduction and discussion. If the authors wish to focus more on the assessment component, I think that is fine, but that should be reflected in the introduction.
Major revisions were undertaken to address this critique in both the introduction and the discussion.
[1] J. A. C. Sterne et al., “Multiple imputation for missing data in epidemiological and clinical research: potential and pitfalls,” BMJ, vol. 338, p. b2393, Jun. 2009, doi: 10.1136/bmj.b2393.
[2] M. J. Silva, Z. Coffee, and C. H. Yu, “Prolonged Cessation of Chronic Opioid Analgesic Therapy: a Multidisciplinary Intervention,” Am. J. Manag. Care, vol. 28, no. 2, pp. 60–65, Feb. 2022, doi: 10.37765/ajmc.2022.88785.
[3] M. J. Silva, Z. Coffee, C. Ho Alex Yu, and M. O. Martel, “Anxiety and Fear Avoidance Beliefs and Behavior May Be Significant Risk Factors for Chronic Opioid Analgesic Therapy Reliance for Patients with Chronic Pain – Results from a Preliminary Study,” Pain Med., no. pnab069, Feb. 2021, doi: 10.1093/pm/pnab069.
[4] S. Z. George, J. M. Fritz, and J. D. Childs, “Investigation of Elevated Fear-Avoidance Beliefs for Patients With Low Back Pain: A Secondary Analysis Involving Patients Enrolled in Physical Therapy Clinical Trials,” J. Orthop. Sports Phys. Ther., vol. 38, no. 2, Art. no. 2, Feb. 2008, doi: 10.2519/jospt.2008.2647.
[5] M. J. Silva, Z. Coffee, Goza, Jessica, and Rumril, Kelly, “Microinduction to Buprenorphine from Methadone for Chronic Pain: Outpatient Protocol with Case Examples,” J Pain Palliat Care Pharmacother, no. 36(1):40-48, Mar. 2022, doi: doi: 10.1080/15360288.2022.2049422.

Round 2
Reviewer 1 Report
The authors did an excellent job with revisions. I have no further suggestions. Thank you for the opportunity to review this manuscript.
Reviewer 2 Report
The authors were attentive to, and adequately addressed, all reviewer comments. I have no further issues to be addressed.